# Transformative Potential and Learning Outcomes of Air Quality Citizen Science Projects in High Schools Using Low-Cost Sensors

**Sonja Grossberndt [1],\*, Antonella Passani [2], Giulia Di Lisio [2], Annelli Janssen [3] and Nuria Castell [1]**

[1] NILU-Norwegian Institute for Air Research, 2007 Kjeller, Norway; ncb@nilu.no
[2] T6 Ecosystems, 00187 Roma, Italy; a.passani@t-6.it (A.P.); g.dilisio@t-6.it (G.D.L.)
[3] The Dutch Research Institute For Transitions (DRIFT), 3062 PA Rotterdam, The Netherlands; janssen@drift.eur.nl
\* Correspondence: sg@nilu.no

**Abstract:** The rise of advanced ICT technologies has made it possible to apply low-cost sensor systems for measuring air quality in citizen science projects, including education. High school students in Norway used these sensor systems in a citizen science project to design, carry out, and evaluate their own research projects on air quality. An impact assessment framework was designed to assess the impact of these activities, considering five areas of impact: scientific, social, economic, political, and environmental. In addition, the framework also considers the transformative potential of the citizen science pilot, i.e., the degree to which the pilot can help to change, alter, or replace current systems, and the business-as-usual in one or more fields such as knowledge production or environmental protection. Data for this assessment were gathered in the form of questionnaires that the students had to complete before starting and after finalizing the pilot activities. The results showed positive impacts on learning, a pro-environmental world view, and an increase in pro-science attitudes and interest in scientific and environmental-related topics at the end of the pilot activities. Only weak impacts were measured for behavioral change. The activities showed transformative potential, which makes the student activities an example of good practice for citizen science activities on air quality with low-cost sensors.

**Keywords:** citizen science; education; low-cost sensor systems; air pollution; learning; transformation

## 1. Introduction

Air quality is one of the major reasons for concern for human health in Europe, resulting in approximately 400,000 premature deaths each year [1]. In order to raise awareness in civil society, and, ultimately, create action towards clean air in Europe, radical changes are needed, which must involve all parts of society [2]. Schools and local communities have a decisive role in this context by promoting knowledge, co-production, and citizen science [3].

Citizen science describes the active involvement of non-professional volunteers in scientific research [4]. These activities have been ongoing for several centuries, although without carrying an explicit name, where nature has been discovered with the help of laymen's observations. Over the last hundred years, scientists have utilized the help of volunteers, especially in environmental observations of astronomy, meteorology, or ornithology [5]. In modern citizen science activities, members of the public can contribute to scientific research at different levels, ranging from project co-design to data collection, analysis of results, and ultimately, their dissemination [6,7]. Participants of citizen science can benefit from these activities by enhancing their scientific literacy and critical thinking, developing new skills, and taking action to influence policy [3,7,8].

Several developments have fostered public participation in scientific activities related to the local environment. The rise of advanced ICT technologies, such as mobile internet

or smartphone applications, and the rapid development within sensing technology, such as low-cost sensor systems for air quality measurements, have facilitated access to data and information and allowed individuals to measure and share with others the state of the environment.

In the last years, we have seen an increased application of low-cost sensor systems in citizen science activities related to air quality (e.g., Sensor.Community (https://sensor.community/en/; accessed on 4 May 2021) and the EU H2020 funded projects ACTION (https://actionproject.eu/; accessed on 4 May 2021), hackAIR (https://www.hackair.eu/; accessed on 4 May 2021) and Smart Citizen (https://smartcitizen.me/; accessed on 4 May 2021)). Citizen science projects have also addressed the issue of air pollution by establishing networks of passive samplers. Examples from Belgium (CurieuzeNeuzen Vlaanderen (https://curieuzeneuzen.be/; accessed on 25 May 2021)) and Italy ("NO$_2$, No Grazie!" (https://www.cittadiniperlaria.org/no2-no-grazie/; accessed on 25 May 2021)) demonstrate that these initiatives have been powerful in terms of attracting the attention of authorities, resulting in changes in air quality management [9].

Regardless of applying active or passive sensors, citizen science activities monitoring air quality have pursued diverse aims, serving both science and the public: (i) providing information about local air quality and potential exposure of the population to air pollution; (ii) awareness campaigns to attract the attention of authorities on issues of air pollution; (iii) complementing the air quality measurements from official monitoring stations, with the aim of improving air quality models; and (iv) improving experience in the use of low-cost measuring devices and their networks [10].

In this paper, we evaluate the learning outcomes from the use of low-cost air quality sensor systems in citizen science activities in high schools. Low-cost sensor systems have the advantage that they are small, easy to handle, and comparably cheap devices for measuring air pollution levels, offering real-time information to their users [10,11]. For this study we selected the air quality sensor Nova SDS011 for particulate matter (PM), which has been previously evaluated in the scientific literature and shown a good agreement when compared against reference data (e.g., [12]). In this way, we ensured that the students engaged in the activity could use the data from the sensors to obtain indicative information about the indoor air quality.

Even though the collected data are not yet of equally good quality as data produced by standardized and certified air quality monitoring devices [13], low-cost systems for air quality are currently being used for many applications, such as creating dense sensor networks for identifying pollution hot-spots [14], complementing measurements taken by official air quality monitoring stations [15], and for educational purposes [16]. Low-cost sensors provide an excellent tool to engage school students in science activities, where they can design their own experiments, monitor their local environment, and gather data that is relevant to their own perceived problem(s) [17]. Participation in citizen science projects moves scientific content from the abstract to the tangible by involving students in hands-on, active learning [18].

The term education can cover different areas, as suggested by [19]. They range from formal education to out-of-school education, local and global communities, families, museums, and online citizen science. In the paper at hand, we focus on formal education.

The activities described in this paper were carried out as a citizen science pilot "Air quality measurements in high schools", funded by the EU H2020 research and innovation project ACTION. We describe the transformative potential and learning outcomes of a citizen science approach in education, which applied low-cost sensors for measuring air quality in Norway. The activities engaged high school students in monitoring air quality in different places in town, their neighborhood, or the indoor environment in their school. In the scientific literature we can find examples of air pollution monitoring in schools conducted by scientists (e.g., [20]). This is a topic of particular interest due to the large timespans that people, and especially school children, spend indoors.

## 2. Materials and Methods

This section summarizes the pilot activities carried out in the Norwegian high schools and describes in detail how their transformative impact and learning outcomes have been assessed.

According to the Norwegian core curriculum for primary and secondary education that was in force during the pilot activities, "school shall allow the pupils to experience the joy of creating, engagement and the urge to explore, and allow them to experience seeing opportunities and transforming ideas into practical actions" [21] (p. 7). The core idea of the citizen science pilot "Air quality measurements in high schools" was to introduce low-cost sensor systems in a high school education context, which would enable the students to actively engage with the topic of air quality. The project activities targeted high schools offering "Teknologi og forskningslære" ("Technology and Science Education") as a school subject. According to its curriculum, this subject aimed at providing basic insight into natural science, technological challenges, and issues in society. It should provide a holistic understanding of technology and natural science being in constant development and the ethical dilemmas connected with this. At the same time, the subject should provide a basis for assessing and discussing technological products and their consequences for society [22].

The citizen science pilot "Air quality measurements in high schools" targeted high schools in the greater Oslo area that were offering the teaching subject "Teknologi og forskningslære" ("Technology and Science Education") in the spring semester of 2020. The activities were designed to align with the needs outlined by the Norwegian Directorate for Education, and thus sparking the interest and involvement of the students.

The pilot activities empowered high school students to design and carry out their own research project about air quality. As preparation for the teachers, who had only a little knowledge about pollution, air quality measurements, and data interpretation, a dedicated teacher workshop was held prior to the student activities. A scientist from the ACTION project consortium that was the coordinator of the citizen science pilot "Air quality measurements in high schools" gave an introductory lecture about those topics. At that workshop, the teachers also spent time in exchanging knowledge and skills with regard to programming Arduino boards and experience from similar projects. The participating teachers invited the scientist to later give a similar lecture to the students prior to the beginning of the citizen science activities. The lecture material was made available to both students and teachers. In these presentations, a special focus was put on explaining the differences in data quality between reference air quality monitoring stations and the low-cost sensor systems the students were going to build and use in their experiments. The students were made aware of the fact that they would obtain only indicative measurement data.

For conducting measurements and collecting data, the students used an Arduino-based air quality sensor platform, equipped with a Nova SDS011 sensor to measure pollutant levels of particulate matter ($PM_{2.5}$ and $PM_{10}$) in the air. They could add additional components for measuring, e.g., relative humidity, temperature, noise, or $CO_2$. The students were in charge of the technical part of mounting and programming the Arduino based sensor, as well as the scientific part of defining the research question, hypothesis, and method, conducting the data analysis, and describing the results and conclusions. The students prepared a scientific poster reporting their experiments and results. They were also obliged to write a report about the activities, which was part of the grading at the end of the semester. Due to the covid-19 pandemic, only three schools participated in the activities as planned. These school classes together built 15 sensors.

The pilot activities underwent an assessment of their impact and transformative potential, which is described in more detail hereinafter.

### 2.1. Framework for Assessing Impact and Transformative Potential

This paragraph describes the framework for assessing the impact and transformative potential as developed within the ACTION project [23], and which was applied to the

citizen science pilot "Air quality measurements in high schools". The framework considers five areas of impact: scientific, social, economic, political, and environmental. Besides these five areas of impact, the framework also considers the transformative potential of the citizen science pilot, i.e., the degree to which the pilot can help to change, alter, or replace current systems, the business-as-usual in one or more fields, such as knowledge production and environmental protection (Figure 1).

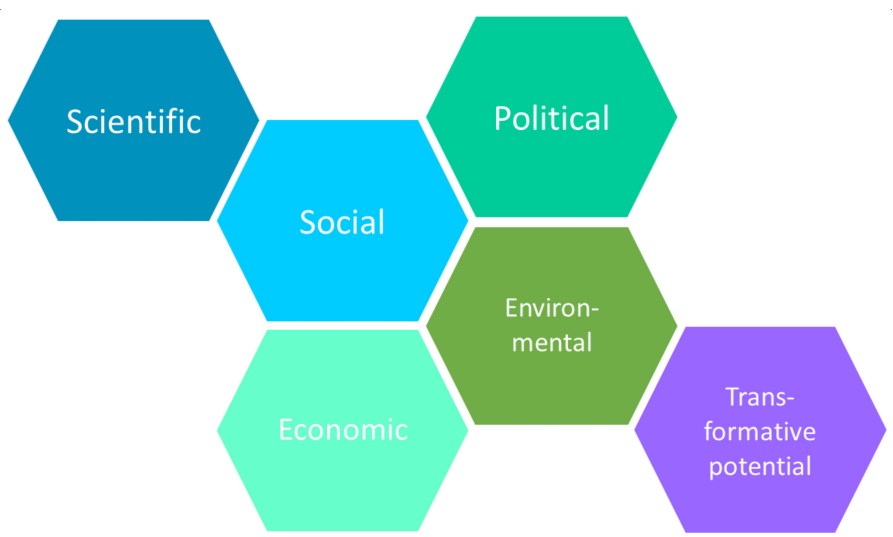

**Figure 1.** Framework for assessing impact and transformative potential.

Each impact area is articulated in several dimensions, for a total of 24 dimensions (Figure 2).

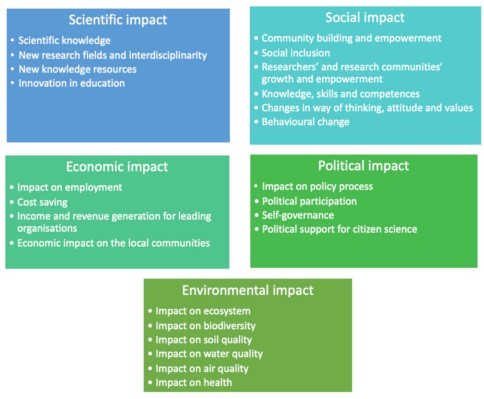

**Figure 2.** ACTION impact assessment areas and dimensions.

The methodology described here is quali-quantitative, following a mixed-method approach. It was designed to be modular and flexible to be adaptable to the specific characteristics of the different citizen science pilots in the ACTION project, but, at the same time, assuring a cross-project and cumulative analysis. Indeed, not all the dimensions are (equally) relevant for all the pilots, depending on their nature, their specific focus, and the level of citizen engagement. The specific needs of a citizen science pilot in terms of impact assessment and the relevance of the various dimensions were collected and presented with the help of an impact assessment canvas [23], a graphic form that supports the managers of each citizen science pilot in mapping their stakeholders, main outputs, and the relevance of the impact dimensions. Then, for each citizen science pilot, an ad hoc impact assessment process was defined, accompanied by the necessary data gathering instruments (questionnaires, focus group guidelines, data recording matrixes). In this

paper, we focus only on the areas of impact that emerged as most relevant for the citizen science pilot "Air quality measurements in high schools", which are described hereafter.

*2.2. Methodology*

The first step of the impact assessment process was defining, together with the project team, the areas of impact and dimensions most relevant for the "Air quality measurements in high schools" pilot. The social dimension emerged as the most relevant, considering the educational nature of the pilot; this area of impact is articulated in six dimensions (Figure 2). For this paper we focus only on the following three of these dimensions:

- Knowledge, skills, and competences;
- Changes in way of thinking, attitude, and values;
- Behavioral changes.

These dimensions emerged as the more promising ones in terms of impact, as they were closer to the pedagogic nature of the pilot compared to the others. To measure the sub-dimensions knowledge, skills and competences, we used three questions inspired by the work of Phillips et al. [24] on evaluation instruments for the learning aspects of citizen science projects. The first questions were: "Have you learnt anything new with this activity?" to which the students had to answer "yes" or "no". If the respondent selected "yes", he/she was asked to describe what he/she learned. Then, two affirmations aimed to understand any changes in interest about the pilot topic (i.e., air quality) and for science more generally. The first one was "Do you think that through the air quality project . . . " and the possible answers were " . . . my interest for this topic has not changed" or " . . . I am more interested in learning about air quality and ways of improving it" or " . . . I am less interested in learning about air quality and ways of improving it". The second one was "After completing the air quality project . . . ", and the possible answers were " . . . my interest for science-related topics and activities has not changed" or " . . . I am more interested in science-related topics and activities" or " . . . I am less interested in science-related topics and activities".

The questionnaires used for the survey, especially for the part related to impacts, relied on self-assessment. In this sense, what we present in this paper is more "the perceived" impact rather than the "objective impact".

In investigating the dimension of changes in way of thinking, attitudes, and values of participants, we examined the project's impact on students' opinions and attitudes towards the topics of air pollution, environment, and towards science. To measure students' opinions on environmental issues we used the New Ecological Paradigm Scale Items (NEPSI), developed in a first version in 1978 by Dunlap and Van Liere [25]. Since then it has been largely used to measure pro-environmental orientation among citizens. The NEPSI was updated more recently [26] to cover a wider range of facets of environmental worldview, measure opinions on exceptionalism and eco crisis facets, and improve some operational aspects. In its last version, NEPSI proposes 15 items mapping 5 hypothesized facets of an ecological worldview (the reality of limits to growth, anti-anthropocentrism, the fragility of nature's balance, the rejection of exemptionalism, and the possibility of an eco-crisis). In the air quality pilot, the students had to indicate to what extend they agreed or disagreed to those 15 items by using a 5-point Likert Scale.

To measure the attitude towards science (related to the dimension "Changes in the way of thinking, attitudes, and values"), we used the (M)ATOSS (modified attitude towards science scale) indicators. The scale measured the participants' predisposition towards the role that science has in society and towards science in general. The scale consisted of four items; the first two items state positive affirmations about science, while the last two underline negative aspects of science. The face validity and content validity of the ATOS and MATOS scales have been assured by the numerous tests carried out at national level for the NSF's science and engineering indicators series [27].

For measuring changes in environmental behavior we used "purchase choices" as a proxy of other pro-environmental behaviors. The term "purchase choices" describes the

choice to buy products that do no or less harm to the environment than other products. For this we used a shorter version of the 30-item Roberts ECCB (Environmental Conscious Consumer Behavior) scale. This scale helped us to measure the changes observed in the participants' behavior with regard to environmentally responsible consumption. Roberts defines the "ecologically conscious consumer" as those who "purchase products and services which they perceive to have a positive (or less negative) impact on the environment" [28] (p. 222). The scale covers a wide range of behaviors and aims to bridge the gap between beliefs and action. Some items were selected from already existing scales, while others were developed by the author to reflect the changes in ecologically conscious consumption in the late 1990s. To predict this behavior in the participants, three independent dimensions were used by the author: the perceived consumer effectiveness (to measure the ability of "individual consumers to affect environmental/resource problems"), the adherence to liberal ideals, and the individual's concern for the environment. To adjust this part of the questionnaire closer to teenagers' consumption experience, we adjusted some of the items and reduced the length of the scale. We also used questions regarding self-perceived efficacy towards the environment. We measured the students' perception of being able to reach the goal of environmental improvement through daily actions, using three simple affirmations, to which the respondents had to agree or disagree on a 5-point Likert-scale. Self-efficacy affects individual decisions, behavior, and persistence in an activity [29–31], so we considered it important to include it in the questionnaires. In this sense being able to increase citizen scientists' self-perceived efficacy can influence their capability to act at a social level in a proactive way, promoting sustainable changes and continuing their involvement in science and learning, and could lead to behavioral change [23].

To measure the respondents' approach to social norms and pressure, they had to answer a question about their propensity to behave pro-environmentally once peers/people in their neighborhood were also engaged in that behavior. Finally, to monitor other potential changes in behavior, the following direct question was added: "Do you think that you will change any of your habits as a result of the activity you carried out?" Students had to answer "yes" or "no". If the selected answer was "yes", the student was asked to describe what he/she was going to change.

The analysis of the transformative potential of the project has its roots in the field of sustainability transitions [32,33], which studies how we can facilitate deep societal changes towards a more sustainable world. Transformative potential "is visible in the extent to which it questions, changes, or challenges (elements of) dominant regimes (e.g., user behavior, technical components, market structures)" (p. 25). We see citizen science as an activity that has the potential to change how science is currently practiced, to be more democratic, open, and citizen led. However, citizen science also has the potential to transform other systems, such as the energy system, mobility system, or problem complexes such as biodiversity, because of the participatory way that citizen science is set up and can thus lead to environmental democracy (https://www.wri.org/blog/2014/07/what-does-environmental-democracy-look; accessed on 4 May 2021).

In order to exploit and assess this transformative potential, we used a framework from the SIC Public Sector Innovation Blog (https://www.silearning.eu/wp-content/uploads/2017/04/6.transformative-impact-tool.pdf; accessed on 4 May 2021) that focuses on five subdimensions, see Table 1 below.

In this table we also see the questions that allowed us to assess these sub-dimensions. These questions were also part of the transformative impact tool: a graphic form which is designed to stimulate learning and critical reflection and in doing so, helps to identify actions and interventions that can increase the transformative impact. As the project manager(s) of a citizen science project have the best overview, the manager of the citizen science pilot "Air quality measurements in high schools" self-evaluated the transformative potential using the impact tool.

**Table 1.** Subdimensions framework to assess transformative potential.

| Subdimensions | Questions Regarding the Project |
|---|---|
| Radical | Is it fundamentally different from dominant practices (in the local context)? Does it "make the impossible possible"? Does it "disrupt the norm"? |
| Iconic | Does it have a "wow effect"? Does it have a communicative, symbolic value? Does it have a clear vision? |
| Catalyzing | Is it appealing/inviting, can people participate and get involved? Does it pave the way for other projects? Could it help break down what is currently the status quo? |
| Timely | Does it play into emerging trends? Are there other initiatives, developments, and actors that can support the project to grow and succeed? |
| Learning | Is it adjustable, scalable, and/or flexible to different contexts and across time? Is there a focus on learning and reflection? |

Figure 3 below visualizes the methods described and applied in this paper with the help of a flow-chart.

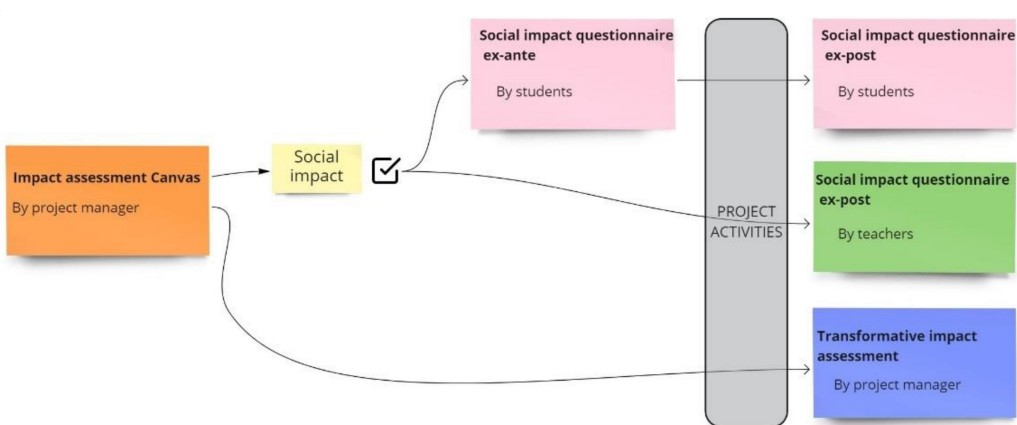

**Figure 3.** Flowchart visualizing the materials and methods for assessing impact and transformative potential of the Norwegian citizen science pilot "Air quality measurements in high schools".

### 2.3. Ex Ante and Ex Post Data Gathering Process and Sampling

Data collection was achieved through two online surveys for the participating students: one at the beginning of the activities, between February and April 2020, and another after the conclusion of the activities, between April and July 2020. The surveys were based on semi-structured questionnaires. Closed questions refer to the scales described in paragraph 2.2, which were complemented by open-ended questions.

The questionnaires collected information about participants in the ex ante phase, evaluated the activities, and measured the impact in the ex post phase. The table below summarizes the categories of information collected through the surveys (Table 2). The transformative potential of the project was assessed by an ex post questionnaire by the citizen science pilot manager.

**Table 2.** Data collected for students, ex ante and ex post.

| Ex Ante Questionnaire | Ex Post Questionnaire |
| --- | --- |
| Demographic characteristics | Demographic characteristics |
| Psychographic characteristics (way of thinking and behaviors towards science and the environment) | Psychographic characteristics (way of thinking and behaviors towards science and the environment) |
| | Evaluation of the activities carried out |
| | Perceived impacts on knowledge and skills |
| | Perceived impact on motivation and interest towards science and air quality |
| | Impact on behaviors |

All questionnaires were translated into Norwegian. For the analysis, the open answers have been translated into English.

We obtained 46 completed ex ante questionnaires and 38 completed ex post questionnaires. In addition, the transformative potential was assessed by the citizen science pilot manager.

## 3. Results

In this section we present the results we obtained from the students through the surveys from the ex ante and the ex post phase, as well as the results obtained from the transformative impact tool. Table 3 displays the students' demographic data.

**Table 3.** Summary of students' demographics.

| | | Ex Ante | | Ex Post | |
| --- | --- | --- | --- | --- | --- |
| | | **N** | **%** | **N** | **%** |
| Age | 16 years | 20 | 44 | 12 | 32 |
| | 17 years | 12 | 26 | 21 | 55 |
| | 18 years | 13 | 28 | 3 | 8 |
| | 19 years | 1 | 2 | | |
| Gender | Male | 28 | 61 | 25 | 66 |
| | Female | 17 | 37 | 11 | 29 |
| | Other | 1 | 2 | | |
| Language spoken at home | Norwegian | 37 | 81 | 32 | 84 |
| | Other | 6 | 13 | 6 | 16 |

As we can see, respondents of the ex ante questionnaire were between 16 and 19 years of age; 20 of them were 16 years old, 12 were 17 years old, and 13 were 18 years old. Only one student was 19 years old.

Respondents of the ex post questionnaire were between 16 and 18 years of age; 12 of them were 16 years old, 21 were 17 years old, and three were 18 years old. We had two missing values for this question.

Between the two phases we registered a difference in the gender distribution. In the ex ante questionnaire 28 respondents (61%) were male, while 17 (37%) were female, and one of the students answered "other". In the ex post questionnaire, the portion of male participants increased to 25 (66%), while the female participants decreased to 11 (29%). We had two missing values. The difference in age and gender distribution was caused by the fact that the surveys were carried out online due to the covid-19 related restrictions and, thus, the teachers could not monitor that all students actually completed the surveys. This resulted in 46 completed questionnaires in the ex ante phase and 38 in the ex post phase.

In order to understand the cultural background of the students without asking about their nationality and without mentioning (for ethical purposes) the migrant/refugee status

they might belong to, the questionnaire included a question about the language used at home. The obtained data in both stages showed that the majority of the students that participated in the pilot (81% for the ex ante phase and 84% for the ex post phase), spoke Norwegian at home, while a minority (13% for the ex ante phase and 16% for the ex post phase) came from families with a cultural background different from Norwegian (two of them were from extra-European countries and two others were from other European countries). For the data gathered before the pilot activities, we registered three missing values here.

In the following, we focus on the dimensions that emerged as most relevant for the citizen science pilot "Air quality measurements in high schools": (1) on the improvement of knowledge and skills; (2) on changes in the way of thinking, attitudes, and values; (3) on behavioral changes; and (4) on the transformative potential of the project.

### 3.1. Knowledge, Skills, and Competences

To measure the impact that the pilot activities had on students' learning, one question was used, with the possibility to give a dichotomic answer ("yes" or "no"). One was "Have you learned anything new with this activity?"

Positive impact was registered for the learning aspect, where 36 out of 38 students affirmed that they had learnt something new with the activities of the citizen science pilot. One value was missing.

To better understand the answers to this question, the students had the possibility to explain their choice. A qualitative analysis of the content of these comments was carried out. The results can be summarized as follows: in general, the students learned more about the scientific and technological aspects than the sociological and humanist ones (as for example, the effects on human health or on society). In fact, 11 students affirmed having gained knowledge about air pollution in general and where it comes from, and 10 students enjoyed the learning experience of building and using sensors. Only six students focused their interest on the consequences of air pollution for human beings and society. Two students enjoyed learning about data analysis, one about programming, and one about actions to reduce pollution. In total, 31 students left a comment, so seven opinions were missing.

To better understand the impact with regard to the learning aspect of the project, students had to complete two sentences with the options provided by the questionnaire. One of those was "Do you think that through the air quality project . . . " and the possible answers were " . . . my interest for this topic has not changed" or " . . . I am more interested in learning about air quality and ways of improving it" or " . . . I am less interested in learning about air quality and ways of improving it". Here an increase in the interest in learning about air quality and ways of improving it was registered for 21 students out of 38. For 14 students, their interest in this topic had not changed. One value was missing. The other sentence was "After completing the air quality project . . . ", and the possible answers were " . . . my interest for science-related topics and activities has not changed" or " . . . I am more interested in science-related topics and activities" or " . . . I am less interested in science-related topics and activities". Here an increased interest in science-related topics and activities was registered for 23 students. For 14 students the interest in science related topics and activities had not changed. One value was missing.

### 3.2. Changes in the Way of Thinking, Attitudes, and Values

During the ex ante phase, 85% of the students declared having a pro-environmental worldview, in the ex post phase this percentage decreased to 78%. At the same time, the percentage of students who seemed to have a non-pro-environmental worldview increased from 5% to 13%. This change is difficult to explain, because the results were not aligned/coherent with the others: so further analysis should be carried out, especially considering the possible difficulties in answering the questions of the New Ecological Paradigm scale. Lack of attention or difficulties in understanding the wording of some items might have influenced the results. This assumption was supported by a student's

comment on the questionnaire during the ex ante phase, stating that the NEP scale items opened "many different interpretations". Another student made a comment about this in the ex post questionnaire, remarking the confusion that he/she felt about the questions in the scale. By analyzing other comments, it could be confirmed that the majority of the students who participated in the pilot activities had in general a pro-environmental attitude and a great respect for nature. To complete the measurement of participants' attitude towards the environment, they had to indicate their degree of agreement or disagreement to the statement "I will behave environmentally-friendly when my friends and family also behave environmentally-friendly". This could help in understanding to what degree the participants were prone to social norms and group pressure or could act as innovators and initiators of new behaviors. A slight improvement was registered between the ex ante and ex post phase, showing an increase in independent attitudes among participants. The students seemed more motivated to act in a pro-environmental way, independently from their family and friends, but the increase (from 76% to 83%) was not particularly great, so further analysis was desirable.

Finally, concerning the changes in view about science, a significant increase in pro-science attitude was registered after the end of the pilot activities in comparison with before the pilot activities (from 83% to 93%).

### 3.3. Behavioural Changes

In the results of the behavioral changes, referring to the green consumption behavior, clear progress in the pro-environmental behavior of respondents was detected. The proportion of participants who affirmed having pro-environmental behavior increased from 52% to 72%, while the "unsure" responses decreased from 41% to 22%, and the proportion of participants engaged in no pro-environmental behavior decreased from 7% to 5%. At the same time, less positive impact has been registered on a direct question of behavioral change. We asked: "Do you think you will change some of your habits as a result of this air quality project?". Only 10 out of 38 students stated that they would change some of their habits as a result of the air quality project, while 26 answered "no" to this qquestion. There were two missing values. From the students who selected "yes" as answer, almost half of them mentioned that they will ventilate their private home. For the other half, two would choose green transport, one would recycle more, one would be more conscious about daily choices, and one would be more conscious to walk, in order to reduce his/her exposure to pollutants by avoiding the most polluted areas.

### 3.4. Transformative Impact

Finally, the manager of the pilot "Air quality measurements in high schools" self-assessed the pilot activities with regard to their transformative impact based on five categories: Radical, Iconic, Catalyzing, Timely, and Learning. The citizen science pilot manager could score each category from 1 (low) to 5 (high). As shown in Figure 4 the pilot "Air quality measurements in high schools" scores 4 on Radical, 3 on Iconic, 3 on Catalyzing, 3 on Timely, and 5 on Learning.

The student pilot scored 4 out of 5 on Radical, because it provided a new education format, away from the traditional front-of-class teaching. The interdisciplinary activities covered different topics, from computer science to technology, science, and society. Designing their own research project and presenting the results in a scientific manner to professional scientists was highly appreciated by the students and brought them closer to science and scientific thinking. Dealing with the topic of air pollution in a self-exploratory manner fostered a holistic view of this scientific and technological topic, with relevance for both, the students themselves and society.

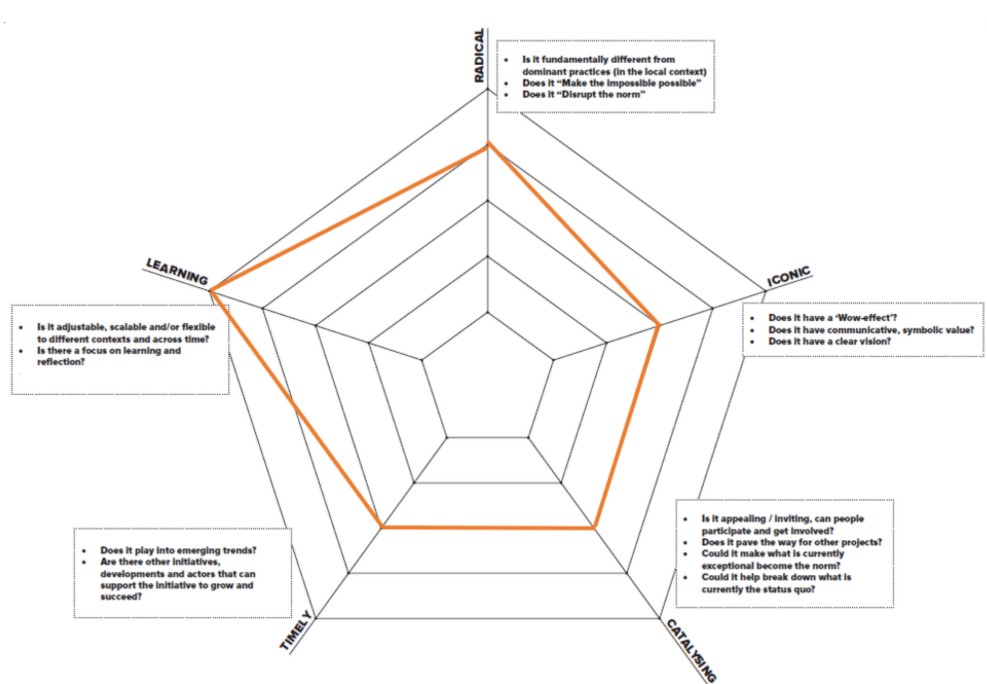

**Figure 4.** Transformative impact of the citizen science pilot "Air quality measurements in high schools" based on the experience of the pilot manager.

The project scored 3 out of 5 on Iconic. While the pilot did not have an enormous "Wow-effect", the activities have been taken up by the teachers positively and feedback from the students was also positive.

With a score of 3 on Catalyzing, there were some ways in which the pilot can inspire other activities that can break down the status quo. The activities have appealed to several teachers who participated with their school classes. The project might not directly "pave the way for other projects", but it might have given teachers new ideas for future education topics, and to contact researchers from other disciplines to carry out projects in other disciplines. This however, requires a lot of initiative by the teachers. The activities will probably also not directly "break down the status quo". This would require changes in the curriculum, which are not always easy to implement. However, it might have been inspirational for teachers to find ways to carry out similar projects within the given frameworks.

The project was quite Timely, scoring 3 out of 5. Focus on the environment is increasing everywhere, and also amongst students. The activities tied in with this development and may have fostered more pro-environmental behavior. They might have also contributed to engaging more female students in technology and research topics/activities.

The highest score was given to the aspect of Learning (5 out of 5). The activities covered different topics that the students could learn about (programming; building sensors; how do sensors work, technology; air pollution, sources, effects on the human body, what can be done to reduce pollution; what are the effects on society, what can we do to avoid emissions to the air; scientific work, also through the poster for the student conference; working independently/exploring topics independently). The activities could be adjusted/upscaled/downsized according to the needs of the students and the frameworks given by the curriculum.

All in all, the pilot scored medium to high on transformative impact, which means the activities contributed to changing the business-as-usual approach, towards a more sustainable world.

## 4. Discussion

Before going into detail about the assessment results, it is important to consider that the impact assessment methodology we applied in this citizen science pilot was based

on self-assessment and indicates perceived changes, not "objective" ones. The perceived impacts and the objective impacts can, of course, differ for many reasons. We aimed to reduce this possible discrepancy by asking the teachers about the impact on students. The results reported by the teachers confirmed the self-assessment of the students [23], nevertheless, the fact that the research design was not a trial-based one with a control group represents a limitation that needs to be considered.

### 4.1. Attitude towards Environment and Science

The majority of the students engaged in the citizen science activities already showed a pro-environmental and pro-science worldview at the beginning of the project. This was coherent with the characteristics of the classes engaged, since they focused on "Technology and science", and thus on the so-called STEM subjects (science, technology, engineering, and mathematics). However, in the ex post analysis we observed an increase in pro-scientific attitudes (in the ex post phase 92% of the students reported pro-science attitudes, while before the beginning of the pilot the percentage was 83%). Many external factors could have been involved in generating this change, but we assume that doing science in a "hands-on" way, carrying out a research activity from the framing of the research question to the dissemination of the results, appears to be a promising path for increasing positive attitudes towards science. Indeed, the majority of the interviewed students declared that after the completion of the pilot activities their interest in science-related topics had increased. For future research it would be interesting to investigate if the process followed in this citizen science pilot could be demonstrated to also be effective in classes without a focus on "technology and science", where students show less interest and appreciation for the environment and science.

### 4.2. From Knowledge Acquisition to Behavioural Change

As reported in the results section, the project achieved positive results in terms of skill acquisition and awareness raising in relation to air pollution. However, the impact on behavioral change was not as evident as the impact on learning, with only a minority of students thinking of changing their behaviors. It is true that the majority of students already showed a positive attitude towards the environment at the beginning of the pilot, so the trigger for changing behaviors was less cogent than it could be in other contexts. In order to better investigate this aspect, we suggested to the teachers to dedicate a specific time and a dedicated activity to this topic in the next round of the pilot. More precisely, we suggested that teachers facilitate a conversation in their classes after the air quality measurement activities, where the students should discuss how they can improve air quality in the areas where the measurement took place and how they could influence other people's behavior in this regard. For future pilot activities, the teachers will be asked to report the main ideas and comments that arose in the open discussion and we will investigate whether this will make any difference to the impact on the students' behaviors. The hypothesis behind this suggestion is that the link between the air quality measurement (the information) and what an individual can do to improve it (behavioral change options) is not that straightforward for students as it would appear, and that a group discussion could support a better framing of such a link, leading to actions.

### 4.3. Transformative Impact and Potential

As described above, the transformative impact of the citizen science pilot was medium to high, which means the activities have the potential to disrupt the status quo. To increase this transformative potential, we suggest two ways forward.

First, the pilot has the potential to change school curricula to make more space for citizen science or other active learning approaches. The pilot cannot do this by itself, but a first step could be to connect the activities to other citizen science projects carried out in high schools in Norway. Forming an (informal) network of these projects could leverage change, in the sense that more schools would see the benefit of citizen science approaches

and could implement them. A step in this direction has already been made by the citizen science pilot manager, by connecting to a national organization that is supporting schools in promoting students' competence within the STEM disciplines. The tutorial for the pilot activities "Air quality measurements in high schools" has already been published on their web pages for free access and download (https://www.lektor2.no/c1336836/nyhet/vis. html?tid=2310436; accessed on 4 May 2021).

Second, the project has the potential to contribute to more awareness of air quality and associated behavioral changes. A promising next step would be to connect to other "niches" (innovative initiatives with the aim of decreasing air pollution) and together forge change in the way we approach air pollution. This change includes social innovation, by activating citizens on the topic, instead of only having the established institutions deal with air pollution. The citizen science approach on air pollution, as well as the young age of the citizen scientists, fits very well with this aspect of social innovation. Another tipping point towards more citizen participation in air quality issues would be to run the project in schools that do not have a science and technology focus. This might not only have more of an effect in terms of attitudes and behavioral change, but might also re-align the gender balance and have a wider reach in general.

## 5. Conclusions

This paper presents a way to use low-cost sensors in an educational context, applying a citizen science approach. The ACTION impact assessment framework has been a useful approach to assess the impact and potential for transformation of the pilot activities in Norwegian high schools. It also supported the pilot team in their reflections on the activities carried out and in acquiring an impact driven approach to citizen science project design.

The analyses revealed that the activities designed for the citizen science pilot "Air quality measurements in high schools" had a positive impact on learning, promoted a pro-environmental world view, increased the interest of young people in scientific and environmental-related topics, and could support pro-science attitudes. It would be interesting to extend the pilot by including other schools, especially those different in terms of socio-cultural background and without a curriculum focusing strongly on STEM. Indeed, it would be interesting to analyze if we would record similar impacts and consider how differences in demographic and/or psychographic variables could influence the impacts. Another path that would be interesting to follow would be implementing a trial base research design and carrying out the impact assessment in classes engaged in the citizen science activities and in classes not engaged (control group), for comparing the results and to better control for external factors that could have influenced this assessment.

The next stage of the citizen science pilot has been slight modified in order to work more on behavioral change: the impact assessment team suggested that the teachers stimulate students in contemplating the links between their everyday behaviors and air quality. In the next phase of the assessment, we will see if this new activity will trigger behavioral changes and to what extent.

Besides the impacts registered amongst the students, the transformative potential of the activities should be emphasized. This makes the pilot activities an example of good practice for citizen science activities on air quality with low-cost sensors in the school context, and we expect to see more activities like the ones described in this paper emerge in other schools in Norway and elsewhere in Europe.

**Author Contributions:** Conceptualization, S.G., A.P. and A.J.; methodology, A.P., A.J.; validation, A.P., G.D.L. and A.J.; formal analysis, A.P., G.D.L. and A.J.; investigation, A.P., G.D.L., and A.J.; resources, A.P., G.D.L. and A.J.; data curation, A.P., G.D.L. and A.J.; writing—original draft preparation, S.G.; writing—review and editing, S.G., A.P., G.D.L., A.J. and N.C.; visualization, A.P., G.D.L., A.J. and S.G.; supervision, A.P., A.J. and S.G.; project administration, S.G., A.P., G.D.L. and A.J.; funding acquisition, S.G., A.P., G.D.L. and A.J. All authors have read and agreed to the published version of the manuscript.

**Funding:** The research leading to this article has been part of the ACTION project. This project has received funding from the European Union's Horizon 2020 research and innovation programme under grant agreement No 824603.

**Institutional Review Board Statement:** The study was conducted according to the guidelines of the Declaration of Helsinki and followed the ethical guidelines and data management plans of the ACTION project (grant agreement No 824603).

**Informed Consent Statement:** Informed consent was obtained from all subjects involved in the study.

**Data Availability Statement:** Questionnaires used for data gathering can be found at the ACTION repository at Zenodo: http://doi.org/10.5281/zenodo.3968460 (accessed on 11 May 2021). The dataset of the impact assessment will be provided during review.

**Acknowledgments:** We would like to thank the students and teachers from the participating high schools in Norway for dedicating their time to contributing to the research that has led to this publication.

**Conflicts of Interest:** The authors declare no conflict of interest.

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
