# Peer review of "Transformative Potential and Learning Outcomes of Air Quality Citizen Science Projects in High Schools Using Low-Cost Sensors"

_atmosphere, doi:10.3390/atmos12060736_

Round 1

Reviewer 1 Report

The work is interesting and well structured. I suggest these additions:

  • in the introductory section, add more detail to the available bibliography (I suggest also citing this work Lucialli et al. "Indoor and outdoor concentrations of benzene, toluene, ethylbenzene and xylene in some Italian schools evaluation of areas with different air pollution" https: //doi.org/10.1016/j.apr.2020.08.007
  • Increase and improve the contents of the introduction, with greater detail on citizen science practices and results, as well as on the use of "low cost" tools for environmental control
  • Line 132: Describe why the 3 dimensions chosen are the most relevant
  • Insert a diagram (e.g. flow chart) that represents the approach described in the materials and methods
  • Insert a table with the summary of the characteristic data of the people who participated in the trial
  • Increase conclusions, including by improving possible future developments 

Author Response

In the introductory section, we have added more detail to the available bibliography, including the suggested reference by Lucialli et al. We further increased the content of the introduction with greater detail on citizen science practices and results, as well as on the use of "low cost" tools for environmental control (lines 51-95).

We provided additional information about why the 3 dimensions chosen are the most relevant for this pilot (lines 180-193)

We inserted a flow chart to visualize the approach described in the materials and methods section (line 281).

We also inserted a table with the summary of the characteristic data of the participants (line 306).

We increased the conclusion section as suggested (lines 534-561).

Reviewer 2 Report

This paper presents a study of the impact of air quality measurements in education.

In general, I find the paper fine and easy to read. This is one of these works that most of the people can understand and is interesting due to its social impact. In this sense, I find this manuscript worth publishing and valuable.

However, some drawbacks can be also said. One of the concerns is that presented results and qualitative but not quantitative. For instance, I would like to see what happens when these experiences are carried out in places with bad air quality and in others with good air quality. Are the outcomes similar or different? Another thing that I usually wonder is about the bias that the type of questions add to the analysis. Many times, outcomes have a bias produced by what the experiments expect to get.

Author Response

We are evaluating the impact on learning and other related social impacts, these are not related to the data they gather (good or bad air quality) but with the process of collecting data so the results will very probably be similar. The intervening variables on social impacts are other, such as the kind of schools, the specific activities carried out etc as it is mentioned in the conclusion section (lines 534-561). 

Regarding bias: the teachers were not “aware” of the impact assessment, i.e., they carried out the activity for pedagogic purposes in the way they planned it. The research design did not influence the activities. As described in the discussion, the assessment we carried out was a self-assessment and indicates perceived changes, not “objective” ones. The perceived impacts and the objective impacts can, of course, differ for many reasons. We aimed to reduce this possible discrepancy by asking the teachers about the impact on students. The results reported by the teachers confirm the self-assessment of the students, nevertheless, the fact that the research design is not a trial-based one with a control group represents a limitation that needs to be considered (lines 464-471).